Improving remote estimation of winter crops gross ecosystem production by inclusion of leaf area index in a spectral model

Juszczak Radosław radoslaw.juszczak@up.poznan.pl 1
Uździcka Bogna 1
Stróżecki Marcin 1
Sakowska Karolina 2
1 Meteorology Department, Poznan University of Life Sciences , Poznań , Poland
2 Institute of Ecology, University of Innsbruck , Innsbruck , Austria
Zhu Zhe
Electronic publication date: 2018 Sep 21
Publication date: 2018
Volume: 6
Electronic Location ID: e5613
Received 2018 Jun 15; Accepted 2018 Aug 21
Copyright: ©2018 Juszczak et al.
Copyright year: 2018
Copyright holder: Juszczak et al.
License: This is an open access article distributed under the terms of the Creative Commons Attribution License, which permits unrestricted use, distribution, reproduction and adaptation in any medium and for any purpose provided that it is properly attributed. For attribution, the original author(s), title, publication source (PeerJ) and either DOI or URL of the article must be cited.
License URL: https://creativecommons.org/licenses/by/4.0/

Keywords: LAI, Spectral vegetation indices, NDVI, SAVI, WDRVI, Gross Ecosystem Production, Croplands, Carbon dioxide fluxes

Funding: Polish Ministry of Science 752/1/N-COST-2 010-0 National Science Centre of Poland 2016/21/B/ST10/02271 European Union’s Horizon 2020 research and innovation programme under the Marie Skłodowska-Curie Grant Agreement 749323 The measurements were funded by the Polish Ministry of Science under project No. 752/1/N-COST-2 010-0 “Assessment of the temporal and spatial variation of the biophysical and spectral indices (NDVI, PRI, WBI) in reference to net exchange of CO2, CH4, H2O between different ecosystems (peatland, forest and arable) and the atmosphere”. During data processing and manuscript writing, Radosław Juszczak was supported by project No. 2016/21/B/ST10/02271 funded by the National Science Centre of Poland. This project has received funding also from the European Union’s Horizon 2020 research and innovation programme under the Marie Skłodowska-Curie Grant Agreement No. 749323. There was no additional external funding received for this study. The funders had no role in study design, data collection and analysis, decision to publish, or preparation of the manuscript.

==============================
The hysteresis of the seasonal relationships between vegetation indices (VIs) and gross ecosystem production (GEP) results in differences between these relationships during vegetative and reproductive phases of plant development cycle and may limit their applicability for estimation of croplands productivity over the entire season. To mitigate this problem and to increase the accuracy of remote sensing-based models for GEP estimation we developed a simple empirical model where greenness-related VIs are multiplied by the leaf area index (LAI). The product of this multiplication has the same seasonality as GEP, and specifically for vegetative periods of winter crops, it allowed the accuracy of GEP estimations to increase and resulted in a significant reduction of the hysteresis of VIs vs. GEP. Our objective was to test the multiyear relationships between VIs and daily GEP in order to develop more general models maintaining reliable performance when applied to years characterized by different climatic conditions. The general model parametrized with NDVI and LAI product allowed to estimate daily GEP of winter and spring crops with an error smaller than 14%, and the rate of GEP over- (for spring barley) or underestimation (for winter crops and potato) was smaller than 25%. The proposed approach may increase the accuracy of crop productivity estimation when greenness VIs are saturating early in the growing season.

Introduction

Leaf area index (LAI) as well as parameters describing carbon dioxide (CO2) exchange between plants and the atmosphere such as net ecosystem production (NEP) and gross ecosystem production (GEP) are key biophysical parameters, which are commonly applied to qualitatively and quantitatively characterize the status of vegetation canopies. Numerous studies have confirmed that based on LAI the intensity of photosynthesis, transpiration, and productivity of plants might be assessed (Borge & Leblanc, 2001; Breda, 2003; Zarco-Tejada, Ustin & Whiting, 2005). Hence, LAI can be used as a proxy of plant growth, biomass and yield, as well as carbon dioxide fluxes exchanged between the ecosystem and the atmosphere (Breda, 2003; Glenn et al., 2008). GEP reflects the total amount of CO2 assimilated by plants in photosynthesis (Waring, Landsberg & Williams, 1998). It depends on the amount, type and physiological condition of plants, but also on climate and habitat conditions (Baldocchi et al., 2001; Keyser et al., 2000). GEP flux analysis is of importance of studies concerning carbon assimilation efficiency at leaf, plant and ecosystem levels. However, due to the limitations of measurement methods, GEP cannot be directly measured in situ. State-of-the-art eddy covariance (EC) systems installed on flux towers make it possible to measure only the net fluxes of CO2 (NEP) (Urbaniak et al., 2016). NEP is defined as a balance of the processes of CO2 exchange between the ecosystem and the atmosphere and is expressed as a difference between GEP and the total amount of CO2 released by the ecosystem to the atmosphere (ecosystem respiration, Reco) (Law et al., 2002).

Remote sensing studies of global vegetation phenology started in 1979 when the meteorological satellite data of Advanced Very High Resolution Radiometer (AVHRR) became available (Goward & Huemmrich, 1992). Since then, the following satellite missions (MODIS, LANDSAT, Sentinel-2) have been providing data which allow remote sensing-based estimation of LAI, fraction of PAR absorbed by plants (fAPAR) and GEP of biomes and ecosystems across the globe with higher spatial and temporal resolutions as well as higher accuracy (e.g., Running et al., 2004; Gao et al., 2017). Traditionally, GEP is estimated as a function of vegetation indices (VI) related to canopy greenness or based on models including in their formulation also Light Use Efficiency (LUE) and/or Photosynthetic Active Radiation (PAR) terms (e.g., Gitelson et al., 2012; Rossini et al., 2012; Sakowska et al., 2014). Many studies have highlighted that simple greenness-related VIs can be successfully used for remote sensing-based estimation of LAI (e.g., Gitelson et al., 2003), chlorophyll content (e.g., Gitelson et al., 2005), fAPAR (Sims et al., 2006), fractional vegetation cover (e.g., Glenn et al., 2008) and GEP (e.g., Prince & Goward, 1995). Among these greenness indices, Normalized Difference Vegetation Index (NDVI) has been the most commonly applied, even though it tends to saturate under conditions of moderate- to high aboveground biomass (e.g., Gitelson, 2004). For this reason, a big effort was undertaken to develop new NDVI-type indices that would not only minimize the soil background influences (e.g., Soil Adjusted Vegetation Index, SAVI, Huete, 1988), but would also overcome the saturation problem in biophysical parameters estimation (Gitelson, 2004). According to existing studies, Modified Simple Ratio (MSR, Chen & Cihlar, 1996), Renormalized Difference Vegetation Index (RDVI, Roujean & Breon, 1995), Wide Dynamic Range Vegetation Index (WDRVI, Gitelson, 2004) or Enhanced Vegetation Index (EVI Huete et al., 2002; Rahman et al., 2005) are more linearly related to fAPAR, LAI, or GEP and they allow to estimate these biophysical parameters with higher accuracy. Due to the limited sensitivity of “greenness” indices to a short-term stress which may not impact the chlorophyll content, the Photochemical Reflectance Index (PRI) was introduced (Gamon, Penuelas & Field, 1992). PRI may be an indicator of the LUE in the process of photosynthesis (Gamon, Penuelas & Field, 1992; Goerner et al., 2011; Penuelas, Filella & Gamon, 1995) and has been used in GEP and LUE estimations at leaf, plant and ecosystem levels (e.g., Rossini et al., 2012; Cheng et al., 2014; Gitelson, Gamon & Solovchenko, 2017).

Both the seasonal dynamics and relationships between VIs and biophysical parameters have been analyzed in numerous studies. The relationships of spectral data have been investigated in relation to LAI (Baret & Guyot, 1991; Law & Waring, 1994; Spanner et al., 1990; Zheng & Moskal, 2009), fAPAR (Asrar et al., 1984; Wang et al., 2004; Sakowska, Juszczak & Gianelle, 2016), and the CO2 fluxes exchanged between the ecosystem and the atmosphere – particularly GEP (Rossini et al., 2012; Sakowska et al., 2014; Skinner, Wylie & Gilmanov, 2011; Uździcka et al., 2017), NEP (Hassan, Bourque & Meng, 2006; Propastin & Kappas, 2009; Veroustraete, Patyn & Myneni, 1996) and NPP (Gower, Kucharik & Norman, 1999; Hunt, 1994; Paruelo et al., 1997; Ruimy, Saugier & Dedieu, 1994). These kind of relationships were mostly analyzed for individual ecosystems (e.g., grasslands - Rossini et al., 2012; Sakowska et al., 2014, peatlands - Chojnicki, 2013, savanna - Sjöström et al., 2009, forests - Xiao et al., 2004), or crop species on a separate basis (e.g., maize - Gitelson et al., 2003; Gitelson et al., 2014; Peng et al., 2011, maize and soybean - Gitelson et al., 2012, rice - Inoue et al., 2008, wheat - Wu et al., 2009).

In this paper we investigated the relationships between LAI, VIs and daily values of GEP determined using chamber measurements conducted on four crops grown in Poland. Similar kind of analyzes where GEP was correlated with VIs, LAI or chlorophyll content, and/or products of VIs and PAR have been presented in many studies (e.g., Rossini et al., 2012; Rossini et al., 2014; Sakowska et al., 2014; Gitelson et al., 2012; Gitelson et al., 2014). The relationships were investigated with an average midday GEP (e.g., Gitelson et al., 2006; Rossini et al., 2012; Sakowska et al., 2014) or with a daily sum of GEP (e.g., Rossini et al., 2012; Gitelson et al., 2014), but in all of these studies GEP was determined based on the ecosystem-scale EC measurements. In this study, GEP was obtained based on plot-scale chamber measurements. Although these measurements have some limitations (e.g., Urbaniak et al., 2016), the biggest advantage of chamber systems is that both NEP and Reco fluxes are measured directly and subsequently which facilitates the calculation of GEP.

In order to obtain remote sensing-based model which allows to estimate daily GEP of crops independently from the type of the crop and climatic conditions with reliable performance, we studied a 3-year dataset (2011–2013) consisting of spectral and biophysical data for two winter (wheat and rye) and two spring (barley, potato) crops. Our specific objectives were to test (1) the accuracy of daily GEP estimations with remote sensing-based models fed with different VIs and VIs*PAR and a model based on LAI, and (2) whether the accuracy of GEP estimations increases when products of VI and LAI are included in the model. Besides, considering that for some crops the hysteresis of the relationships between VIs and biophysical parameters is observed (Peng et al., 2017), we aimed at developing a simple linear empirical model based on VIs and LAI in order to reduce hysteresis of the VIs vs. GEP relationships between vegetative and reproductive phases of crop development cycle and to increase the accuracy of daily GEP (GEPd) estimations of croplands. According to our knowledge this is also the first study in which CO2 fluxes measured with chambers are combined with both LAI and VIs.

Material and Methods

Experimental site

Measurements were conducted at the Brody Experimental Station (52°26′N, 16°18′E) on plots of the long-term experiment that has been conducted since 1957 by the Department of Agronomy, Poznań University of Life Sciences, Poland (Blecharczyk et al., 2016). Crops were grown in the crop rotation and monoculture systems under 11 different fertilization regimes (no fertilization, manure, manure + NPK, NPK + Ca-CaO, NPK, NP, NK, PK, N, P, K). The measurements presented in this paper were performed on four crop species: potato (var. Wineta), spring barley (var. Nadek), winter wheat (var. Turkis) and winter rye (var. Dankowskie Zlote), grown in a seven-year rotation (potato → spring barley → winter triticale →1- and 2-year alfalfa → winter wheat → winter rye). The investigated crops were fertilized with NPK (90 kg N ha−1 a−1, 60 kg P2O5 ha−1 a−1, 120 kg K2O ha−1 a−1) with an addition of Ca-CaO (1.5 Mg CaO ha−1 a−1) and grown in 6 × 11 m plots separated by 0.5 m wide bare soil stripe. The annual mean air temperature of the study area is 7.9 °C, while the annual precipitation sum is 571 mm (average for 1959–1999). The soils are classified as Albic Luvisols developed on loamy sands overlying loamy material (Majchrzak et al., 2016).

Chamber CO2 fluxes and flux modelling

Measurements of CO2 fluxes (NEP, Reco) were taken using a closed dynamic portable chamber system, which consisted of a transparent and non-transparent chambers to measure NEP and Reco fluxes, respectively (Chojnicki et al., 2010; Juszczak et al., 2013). Measurements were made in two subplots for each crop. The transparent chamber was made of three mm-thick Plexiglas (Evonik Industries, Darmstadt, Germany), as this material has a high solar radiation transmittance (approximately 90%, Acosta et al., 2017; Hoffman et al., 2015). The non-transparent chamber was made of three mm-thick white PVC to ensure dark conditions inside the chamber. The chambers had dimensions of 0.78 × 0.78 × 0.50 m and a total volume of 0.296 m3. In case of winter rye and winter wheat, the extensions of 0.5 m height were used (made of the same material) in order to adopt the height of chamber to the height of the canopy. During the measurements, chambers were placed on square PVC collars (0.75 × 0.75 m), inserted into the soil just after sowing. The insertion depth of the collars was 15 cm. The chambers were equipped with a set of computer fans (1.4 W; 1,500 rpm each) mixing the air and a vent to equilibrate pressure in the chamber headspace. The air temperature inside the chamber headspace was measured with a radiation-shielded thermistor (T-107; Campbell Scientific, Logan, UT, USA) at a height of 0.3 m, or 0.8 m for short and tall chambers, respectively (Juszczak, Acosta & Olejnik, 2012; Juszczak et al., 2013). The CO2 concentration changes in the chamber was measured using LI-820 gas analyzer (LI-COR Inc., Lincoln, NE, USA). The air was circulated between the chamber and the analyzer in a closed loop with the flow rate of 0.7 l min−1. In order to keep the air temperature inside the chamber headspace stable, the transparent chamber was cooled with a passive system described in Acosta et al. (2017).

Chamber measurements were taken every 3–5 weeks throughout the entire year (including winter) on cloudless days from sunrise to late afternoon (Uździcka et al., 2017). However, when any clouds appeared (usually in the afternoon) a particular attention was paid to perform measurements at stable PAR conditions. Overall, 37 chamber campaigns were conducted in the years 2011–2013. Measurements of NEP and Reco were taken on each of the soil frames several times per day (from five to 12, depending on the daytime length). A single NEP measurement took 120 seconds, and the subsequent NEP measurement at the same plot was taken before the value of incoming PAR changed by more than 150 µmol m−2 s−1. A single measurement of Reco took 180 seconds and the succeeding measurements were taken before the soil temperature at the same plot changed by more than 0.5 °C. The CO2 flux (F) in µmols m−2 per time unit (t) was calculated from the gas concentration change in the chamber headspace ΔCΔt, the chamber volume (V) and the enclosed soil area (A) from the following equation (Urbaniak et al., 2016): (1) F=ΔCΔt⋅VA⋅Mv

where Mv (m3 mol−1) is the molar volume of air at a given chamber air temperature and pressure. The determination coefficient (r2) was calculated for each single chamber closure time and if r2 < 0.8, the fluxes were excluded from the analyzes. To ensure that good quality near-zero fluxes were not erroneously excluded by this criteria, all rejected fluxes were visually inspected.

Fluxes of NEP, Reco and GEP for measurement days and periods between campaigns were calculated using a simple empirical model described by Drösler (2005) and farther elaborated by Hoffman et al. (2015). For this purpose, for each day of measurements, the relationships between measured Reco and temperature were established by fitting to the campaign-specific flux dataset the temperature dependent Arrhenius-type respiration model of Lloyd & Taylor (1994). Using the parameters of this model, Reco values at the time of the NEP measurements were estimated based on the measured temperatures (Juszczak et al., 2013). In the next step, based on such estimated Reco and measured NEP, GEP was calculated according to the formula GEP = NEP + Reco. Subsequently, the calculated GEP fluxes were correlated with measured PAR, fitting to the campaign-specific GEP dataset a rectangular hyperbolic light response Michaelis–Menten kinetic model (Michaelis & Menten, 1913). The Reco and GEP model parameters were interpolated linearly for the periods between the campaigns with a 30-minute step, so that, based on the continuous time series of measured PAR and temperature (means for 30-minute periods), GEP and Reco was calculated. NEP was calculated from the formula NEP =GEP − Reco. Daily sums of GEP (so called daily GEP (GEPd)) were calculated as a sum of all 30-minute GEP fluxes estimated for each day with the flux model in between sunrise to sunset.

Measurements of LAI and spectral characteristics of the crops

LAI and multispectral data were collected during the growing season, from March to October, at one- to two-week intervals. At the spring barley plots measurements started in the middle of April (after sowing), whereas in case of potatoes in the 2nd week of May (after planting). Only the data collected in the period between April and the 2nd week of August (just after harvesting) was considered in the analyzes, hence we did not present nor analyzed data collected after sowing of winter crops and after harvest. The dates of the measurement campaigns were selected so that these measurements could overlap with chamber measurements of CO2 exchange. However, when the weather conditions were not stable due to appearing clouds, the measurements were repeated on the first sunny day following the chamber measurements. 36 measurement campaigns were organized in the 3-year period (2011–2013). Spectral measurements were carried out only on sunny days, always around the solar noon (between 10:00–14:00). Measurements of LAI and reflectance were taken at each plot in three replications, always in the same locations.

LAI was measured by means of the SunScan system (Delta-T Devices, Cambridge, UK). The spectral characteristics of the surface of plant-covered plots were measured using two 4-channel SKR1850 sensors (SKYE Instruments Ltd., Llandrindod Wells, UK) mounted on a portable SKL908 device (Spectrosense2+). Incident and reflected radiation was recorded at central wavelengths of 531, 570, 670 and 850 nm with 10 nm bandwidths. Next, applying the methodology developed by SKYE instruments Ltd. (SpectroSense2+ Manual), vegetation indices (NDVI, SAVI and PRI) were calculated (Table 1) according to the formula: VI=Z⋅R1r⋅Y−R2r⋅XZ⋅R1r⋅Y+R2r⋅X

where VI is a vegetation index, Z is a ratio sensitivity of reflected R1r:R2r; X and Y are incident readings for R1i and R2i, respectively (in µmol m−2s−1), while R1r and R2r correspond to reflected signal readings for wavelengths R1 and R2 (in nanoamps). For NDVI and SAVI, R1 and R2 correspond to 850 nm and 670 nm wavelengths respectively, while for PRI they correspond to wavelengths 570 nm and 531 nm. To calculate SAVI, the above formula was modified according to equation provided in Table 1. Wide Dynamic Range Vegetation Index was calculated from NDVI from the equation: WDRVI = [(α + 1) NDVI + (α − 1)]/[(α − 1)NDVI + (α + 1)], where α = 0.2 (Vina & Gitelson, 2005). In order to express values of this VI in positive numbers, sWDRVI was calculated from equation (sWDRVI =(WDRVI + 1)∕2).

Table 1 Spectral vegetation indices calculated from ground-based spectroscopy and presented in this study.

Spectral vegetation index	Formulation	Reference	
NDVI	NDVI=ρ850−ρ670ρ850+ρ670	Rouse et al. (1973)	
SAVI	SAVI=ρ850−ρ670ρ850+ρ670+L1+L	Huete (1988)	
PRI	PRI=ρ570−ρ531ρ570+ρ531	Gamon, Penuelas & Field (1992)	
WDRVI	WDRVI=α∗ρ850−ρ670α∗ρ850−ρ670	Gitelson (2004)	
Notes.

ρ reflectance at a given wavelength

NDVI Normalized Difference Vegetation Index

SAVI Soil Adjusted Vegetation Index

PRI Photochemical Reflectance Index

WDRVI Wide Dynamic Range Vegetation Index

Models for GEP estimations

Daily GEP (GEPd) were estimated based on linear regressions assuming a direct linear relationship between GEPd and LAI (model 1), GEPd and VIs (model 2), GEPd and a product of VIs and mean daily PAR calculated for the time between the sunrise and sunset −PARd (model 3), as well as GEPd and a product of VIs and LAI (model 4) (Table 2). All the models were tested based on the combined multi-year (2011–2013) dataset for each crop species separately, as well as for all cereals: winter wheat, winter rye and spring barley (i) and cereals and potatoes (ii) considered together in order to develop the general models for GEPd estimations for croplands. The general models with the best goodness of fit were then tested for each crop independently.

Table 2 The models tested for GEPd estimation in the present study.

Model	Model formulation	
1	GEPd = a⋅LAI + b	
2	GEPd = a⋅VI + b	
3	GEPd = a⋅(VI⋅PARd) + b	
4	GEPd = a⋅(VI⋅LAI) + b	

Statistical analysis

Pearson’s correlation analysis was used to test the significance of the relationships between GEPd and (i) LAI; (ii) VIs (NDVI, SAVI, WDRVI, PRI); (iii) VIs*PARd; and (iv) VIs* LAI. Analyzes were conducted for each crop independently and for the combined datasets of cereals, as well as for cereals and potatoes considered together.

Each of the four analyzed models’ coefficients were found by fitting each model against GEPd. Goodness of fit statistics (coefficient of determination, R2; root means square error, RMSE in gCO2-C m−2d−1; and normalized root mean square error, NRMSE in %) were computed to compare the performance of the models.

In order to determine if there are significant differences in relationships between GEPd and VIs between the vegetative and reproductive phases of crop development, the two-sample t-test approach was applied. The differences between analyzed relationships were considered to be significant if p-value obtained from the test was lower than 0.05.

Due to limited number of data (n equals from 21 to 26, depending on the crop and VI), validation of the best performing general models developed based on the combined dataset for cereals and potatoes was conducted for each crop species separately by comparing GEPd retrieved from chamber CO2 fluxes with GEPd estimated based on the spectral model individually for each crop.

Results

The seasonal variations of meteorological conditions for the analyzed growing seasons are presented in Fig. 1. The seasonal mean air temperatures were 14.0 °C, 13.9 °C and 12.4 °C, while the sums of precipitation were 331 mm, 363 mm and 344 mm for the periods between 1st of March and 31st of August of 2011, 2012 and 2013, respectively. The highest amount of precipitation was recorded in July 2011 (179 mm), July 2012 (99 mm), and June 2013 (125 mm). It has to be highlighted that sums of precipitation in May and June of 2011 - the two most critical for the plant growth months - were nearly two-times smaller than during the same period of the two following years. Considering the long term climatological records for this region, 2011 and 2013 are considered as warm (with the mean annual temperatures of 9.4 °C and 8.7 °C, respectively) and dry years (507 mm and 503 mm, respectively), while 2012 was considered as a warm (with the mean annual temperature of 9.0 °C) and wet (592 mm) year (Uździcka et al., 2017). During the growing period of the main crop (March-August) the average PARd was 659 (±241), 664 (±224) and 641 (±223) µmol m−2s−1 in 2011, 2012 and 2013, respectively, with the maximum values of 1,011–1,068 µmol m−2s−1 (Fig. 2A).

Figure 1 Monthly sums of precipitation and mean air temperatures in Brody for 2011–2013.

Figure 2 Seasonal variations of mean daily PAR (PARd) at the study site and daily GEP (GEPd; circles) and LAI (triangles) for the analyzed crops in the years 2011–2013.

(A) mean daily PAR; (B) refers to winter wheat; (C) winter rye; (D) spring barley and (E) potatoes. Shaded areas indicate periods when the main crop was present in the field (these periods were analyzed in this study).

The seasonal variations of LAI and GEPd for all analyzed years and crops are presented in Fig. 2. The maximum LAI (LAImax) and GEPd of winter crops were recorded in the first week of May in 2011 and the last week of May-beginning of June in 2012 and 2013. In case of spring barley the LAI and GEPd peaks occurred during the last week of May in 2011 and 2012 and in mid-June in 2013. Maximum rates of GEPd and LAI at potato plots were recorded in the 2nd week of June in 2013 and 3rd week of June in 2011 and 2012. Maximum values of GEPd and LAI for winter wheat were observed nearly at the same time, with just a 2-week shift in 2011. The LAImax was 2.7, 3.3, and 4.4 m2m−2 for winter wheat; 2.9, 5.0, and 4.3 m2m−2 for winter rye; 2.5, 4.0 and 4.0 m2m−2 for spring barley and 1.8, 3.4 and 3.2 m2m−2 for potato, in 2011, 2012 and 2013, respectively (Table 3). The comparison of LAImax between the investigated years showed that for all the investigated crops LAImax was lower in 2011 than in both 2012 and 2013. There were no differences in LAImax for spring barley and potato in 2012 and 2013, while LAImax was the highest for winter rye in 2012 and for winter wheat in 2013.

Table 3 Seasonal mean and maximum values of GEPd, NDVI, SAVI, sWDRVI, PRI), canopy height (Hcanopy) and fractional cover (F%) for the analyzed crops in the growing periods of 2011, 2012 and 2013.

		Units	Winter wheat	Winter rye	Spring barley	Potatoes	
			mean	max.	mean	max.	mean	max.	mean	max.	
	GEPd	gCO2-Cm−2 s−1	3.60(±3.4)	12.14	6.31(±4.3)	14.30	3.75(±2.7)	11.33	2.77(±3.1)	10.54	
	LAI	m2 m−2	1.56(±0.7)	2.67	2.14(±0.6)	2.87	1.19(±1.0)	2.53	0.83(±0.8)	1.80	
	Hcanopy	m	0.41(±0.2)	0.78	0.78(±0.6)	1.50	0.22(±0.2)	0.62	0.15(±0.2)	0.55	
2011	F%	%	0.38(±0.2)	0.60	0.66(±0.2)	0.90	0.37(±0.3)	0.85	0.19(±0.3)	0.70	
	NDVI	–	0.65(±0.2)	0.79	0.68(±0.2)	0.88	0.51(±0.3)	0.85	0.23(±0.3)	0.77	
	SAVI	–	0.17(±0.1)	0.28	0.18(±0.1)	0.33	0.16(±0.1)	0.30	0.08(±0.1)	0.28	
	sWDRVI	–	0.47(±0.2)	0.63	0.55(±0.2)	0.75	0.37(±0.2)	0.71	0.37(±0.2)	0.61	
	PRI	–	−0.19(±0.02)	−0.18	−0.18(±0.02)	−0.15	−0.20(±0.02)	−0.17	−0.19(±0.01)	−0.18	
2012	GEPd	gCO2-Cm−2 s−1	7.24(±5.9)	20.33	8.78(±7.2)	31.65	5.78(±4.5)	14.36	6.31(±3.2)	12.66	
LAI	m2 m−2	1.73(±1.15)	3.33	3.12(±1.6)	5.00	2.05(±1.3)	4.03	1.84(±1.3)	3.37	
Hcanopy	m	0.54(±0.4)	0.99	0.95(±0.6)	1.59	0.34(±0.3)	0.66	0.30(±0.3)	0.72	
F%	%	0.41(±0.3)	0.80	0.75(±0.3)	0.95	0.56(±0.4)	0.95	0.39(±0.4)	0.95	
NDVI	–	0.48(±0.3)	0.84	0.56(±0.3)	0.90	0.44(±0.3)	0.84	0.55(±0.3)	0.88	
SAVI	–	0.15(±0.1)	0.31	0.18(±0.1)	0.31	0.17(±0.2)	0.38	0.22(±0.2)	0.49	
sWDRVI	–	0.39(±0.2)	0.70	0.49(±0.2)	0.79	0.35(±0.2)	0.69	0.46(±0.2)	0.77	
PRI	–	−0.19(±0.01)	−0.17	−0.20(±0.03)	−0.16	−0.22(±0.02)	−0.20	−0.20(±0.02)	−0.23	
2013	GEPd	gCO2-Cm−2 s−1	6.91(±6.3)	22.24	8.13(±6.3)	27.29	5.37(±4.2)	14.06	5.66(±3.6)	13.33	
LAI	m2 m−2	2.33(±1.8)	4.43	2.66(±1.8)	4.33	1.81(±1.5)	4.02	1.13(±1.2)	3.23	
Hcanopy	m	0.38(±0.3)	0.82	0.80(±0.6)	1.40	0.28(±0.2)	0.60	0.24(±0.2)	0.45	
F%	%	0.49(±0.4)	0.90	0.62(±0.3)	0.90	0.42(±0.4)	0.90	0.30(±0.2)	0.50	
NDVI	–	0.58(±0.4)	0.88	0.55(±0.3)	0.85	0.57(±0.3)	0.89	0.47(±0.3)	0.70	
SAVI	–	0.25(±0.2)	0.44	0.24(±0.1)	0.38	0.33(±0.1)	0.45	0.22(±0.1)	0.30	
sWDRVI	–	0.58(±0.2)	0.76	0.46(±0.2)	0.72	0.44(±0.2)	0.78	0.38(±0.2)	0.56	
PRI	–	−0.19(±0.03)	−0.17	−0.20(±0.03)	−0.16	−0.17(±0.1)	0.00	−0.20(±0.01)	−0.19	
Notes.

For winter crops growing periods started on the 1st of March; for spring barley they started in the middle of April and lasted until skimming in the last week of August; for potato the growing seasons started in the second week of May and ended in the middle of September each year.

Similarly to LAI, the maximum daily GEP values of all the investigated crops were lower in 2011 compared to 2012 and 2013. The GEPd of winter crops was 45% (winter rye) to 60% (winter wheat) lower in 2011 than in the two following years. The differences in maximum GEPd of the spring crops (spring barley and potato) between 2011 and 2012, 2013 were smaller than in case of winter crops (20%). Maximum rates of GEPd for winter wheat reached 12.1, 20.3 and 22.2 gCO2-Cm−2 d−1 and for winter rye 14.3, 31.6 and 27.3 gCO2-Cm−2 d−1 in 2011, 2012 and 2013, respectively, while for spring barley and potato GEPd were 11.3 and 10.5 gCO2-Cm−2 d−1 in 2011, 14.4 and 12.3 gCO2-Cm−2 d−1 in 2012, and 14.0 and 13.3 gCO2-Cm−2 d−1 in 2013, respectively (Table 3).

Maximum values of NDVI and SAVI were observed nearly at the same time as the peaks of LAI and GEPd (data shown in File S2). Moreover they were also lower in 2011 than in the other two years, even though the observed differences in maximum VIs values were less prominent than the differences in the analyzed biophysical parameters (Table 3). All these data clearly indicate the effect of drought which occurred in the late spring - early summer of 2011, when sums of precipitation were 50% smaller than the precipitation observed in the same period of 2012 and 2013.

The analysis of linear regression revealed that LAI explained minimum 60% of the variability in GEPd (Table 4). NDVI and SAVI explained from 52% to 72% of the variability in GEPd for winter crops and up to 81%-91% for spring crops, when crops were considered separately. For crop-combined dataset, which consisted of the data of all the analyzed crops, NDVI and SAVI explained 50% to 65% of the variability in GEPd. SAVI-based models worked better only in case of models developed for winter rye, while in case of other crops it did not lead to more accurate estimations of GEPd. Moreover, the SAVI-based models (R2 = 0.50, NRMSE = 18.24%) developed for all the crops together were less accurate than NDVI-based models (R2 = 0.65, NRMSE = 15.29%). The sWDRVI, which was expected to be more linearly correlated with GEPd, explained between 50% (winter rye) to 86% (spring barley) of the GEPd variability and did not improve the accuracy of GEPd estimations in the crop-combined model (R2 = 0.65, NRMSE = 15.60%). The highest accuracy of model 2 was obtained for potato (if based on NDVI; R2 = 0.91, NRMSE = 10.94%). Inclusion of PARd into model 3 did not improve estimations of GEPd for any of the investigated crops nor for the crop-combined datasets.

Table 4 Summary of the statistics of linear regressions between LAI, VIs and GEPd for each crop individually, for all cereals (winter and spring crops) and for cereals and potato grouped together.

Model		Winter wheat	Winter rye	
		n	R2	RMSE	NRMSE	p	n	R2	RMSE	NRMSE	p	
		–	–	gCO2-C m−2 s−1	%	–		–	gCO2-C m−2 s−1	%	–	
1	LAI	26	0.62	4.07	18.75	<0.0001	26	0.60	4.34	18.54	<0.0001	
2	NDVI	25	0.56	4.31	19.83	<0.0001	26	0.52	4.72	20.15	<0.0001	
SAVI	21	0.59	3.93	18.56	<0.0001	24	0.72	3.53	15.07	<0.0001	
sWDRVI	25	0.62	4.03	18.54	<0.0001	26	0.50	4.84	20.67	<0.0001	
PRI	22	0.49	4.64	19.79	<0.001	21	0.49	4.63	19.79	<0.0001	
3	NDVI*PARd	25	0.54	4.40	20.24	<0.0001	26	0.59	4.39	18.76	<0.0001	
SAVI*PARd	21	0.52	4.29	20.25	<0.0001	24	0.57	4.38	18.73	<0.0001	
sWDRVI*PARd	25	0.59	4.21	19.40	<0.0001	26	0.58	4.41	18.85	<0.0001	
4	NDVI*LAI	25	0.70	3.56	16.38	<0.0001	26	0.79	3.14	13.42	<0.0001	
SAVI*LAI	21	0.63	3.81	17.67	<0.0001	24	0.80	3.01	12.86	<0.0001	
sWDRVI*LAI	25	0.69	3.61	16.61	<0.0001	26	0.73	3.55	15.16	<0.0001	
		Spring barley	Potatoes	
1	LAI	22	0.65	2.51	17.87	<0.0001	22	0.75	2.18	18.99	<0.0001	
2	NDVI	25	0.82	1.73	12.33	<0.0001	22	0.91	1.26	10.94	<0.0001	
SAVI	23	0.81	1.77	12.63	<0.0001	19	0.90	1.40	12.21	<0.0001	
sWDRVI	25	0.86	1.55	10.99	<0.0001	22	0.85	1.66	14.45	<0.0001	
PRI	21	0.05	3.88	27.24	0.235	21	0.03	4.22	36.75	0.4533	
3	NDVI*PARd	25	0.82	1.70	12.12	<0.0001	22	0.86	1.54	13.43	<0.0001	
SAVI*PARd	23	0.83	1.66	11.81	<0.0001	19	0.81	1.64	14.26	<0.0001	
sWDRVI*PARd	25	0.84	1.64	11.69	<0.0001	22	0.79	1.87	16.25	<0.0001	
4	NDVI*LAI	22	0.83	1.79	12.70	<0.0001	22	0.76	2.10	18.30	<0.0001	
SAVI*LAI	22	0.78	1.92	13.67	<0.0001	22	0.76	1.99	17.30	<0.0001	
sWDRVI*LAI	22	0.81	1.73	12.80	<0.0001	22	0.75	2.12	18.49	<0.0001	
		ALL cereals		Cereals + potatoes	
1	LAI	74	0.61	4.10	17.09	<0.0001	96	0.65	3.79	15.81	<0.0001	
2	NDVI	76	0.60	4.09	17.02	<0.0001	98	0.65	3.67	15.29	<0.0001	
SAVI	68	0.54	4.34	18.09	<0.0001	87	0.50	4.38	18.24	<0.0001	
sWDRVI	76	0.62	4.03	16.79	<0.0001	98	0.65	3.74	15.60	<0.0001	
PRI	64	0.37	4.86	20.26	<0.0001	85	0.29	4.98	20.73	<0.0001	
3	NDVI*PARd	76	0.59	4.10	17.10	<0.0001	98	0.61	3.79	15.79	<0.0001	
SAVI*PARd	68	0.48	4.61	19.19	<0.0001	87	0.39	4.76	19.85	<0.0001	
sWDRVI*PARd	76	0.58	4.15	17.30	<0.0001	98	0.59	3.88	16.16	<0.0001	
4	NDVI*LAI	73	0.74	3.42	14.39	<0.0001	95	0.74	3.26	13.57	<0.0001	
SAVI*LAI	67	0.63	3.87	16.12	<0.0001	89	0.56	4.03	16.79	<0.0001	
sWDRVI*LAI	73	0.65	3.88	16.18	<0.0001	99	0.71	3.40	14.16	<0.0001	
Notes.

n number of observations

R2 coefficient of determination

RMSE root mean square error

NRMSE normalized root mean square error

The best performing models are in bold print.

The inclusion of LAI into the “VIs”-based models resulted in a general increase of their performance in case of the winter crops (Table 4). The highest increase of the accuracy of GEPd estimations was found for NDVI-based models. For winter wheat, RMSE decreased from 4.31 to 3.56 gCO2-C m−2 d−1, while NRMSE decreased from 19.83% to 16.38% after inclusion of LAI into NDVI-based model. For winter rye, the changes were even more prominent - RMSE decreased from 4.72 to 3.14 gCO2-C m−2 d−1, whereas NRMSE decreased from 20.15% to 13.42%. Inclusion of LAI into NDVI-based models developed for spring barley and potatoes led to a decrease of the accuracy of GEPd estimations (Table 4). For spring barley RMSE and NRMSE increased from 1.73 to 1.79 gCO2-C m−2 d−1 and from 12.33% to 12.70%, respectively. The highest reduction of the model accuracy was observed for potatoes - RMSE and NRMSE increased from 1.26 to 2.10 gCO2-C m−2 d−1 and from 10.94% to 18.30%, respectively.

For more general models developed for the cereals and crop-combined datasets the inclusion of LAI into the VIs-based models also resulted in an improvement of model performance. RMSE and NRMSE of NDVI and LAI-based models were the smallest and decreased from 4.09 to 3.42 gCO2-C m−2 d−1 and from 17.02% to 14.39% for cereals-combined models, and from 3.67 to 3.26 gCO2-C m−2 d−1 and from 15.29% to 13.57% for crop-combined datasets, respectively (Table 4). That is why we used the most general and the best fitting NDVI*LAI model developed for all the crops together (cereals & potatoes, Fig. 3) to estimate GEPd values for each crop separately (Fig. 4). Although there is a good agreement between observed and predicted GEPd values for all the crops (Fig. 3), GEPd estimated for winter crops and potatoes was underestimated, while GEPd of spring barley was overestimated, but the rate of over- or under-estimation did not exceed 25% (Fig. 4).

Figure 3 Scatterplots of relationships between (A) NDVI*LAI and GEPd and (B) observed and predicted GEPd estimated for all the crops considered together, based on the general crop-combined model.

Figure 4 Scatter plots of relationships between observed and predicted GEPd for the analyzed crops.

(A) winter wheat; (B) winter rye; (C) spring barley and (D) potatoes. GEPd for each crop was estimated based on the general crop-combined model: GEPd = 4.87∗(NDVI∗LAI) + 0.61.

Discussion

Most of the remote sensing-based models to estimate GEP of croplands are rather crop-specific. They were developed for maize, soybean (Gitelson et al., 2012), rice (Inoue et al., 2008), wheat (Wu et al., 2009), rye, barley, potato (Uździcka et al., 2017) and in the majority of studies it has not been tested if these models can be directly applied (without reparametrization) for estimation of CO2 uptake for other crops. Here we presented a more general and robust approach based on combining the datasets for the two winter and two spring crops together and we proved that the accuracy of crop-combined models is not different from those developed for winter crops on a separate basis (NRMSE of the best crop-combined models are in range of 13–16%), but it may be lowered compared to simple VI-based models developed for spring barley and potato (NRMSE is in range of about 11% for spring crops, whereas NRMSE of the best crop-combined model is 13.57%, Table 4).

The weakness of our study is related to the applied method for spectral properties measurements. The Spectrosense 2+ measuring system with 4-channel upward- and downward-looking multispectral radiometers allows to measure incident and reflected radiation only in four most commonly used bands (NIR, RED and GREEN wavelengths). Due to missing measurements at red-edge, blue and other specific wavelengths we were not able to calculate many other important indices more sensitive to medium to high biomass (e.g., NDVIred–edge (Gitelson & Merzlyak, 1994), Normalized Difference Structural Index, NDSI (Vescovo et al., 2012)) or EVI (Huete et al., 2002), which could help to overcome the saturation effect of classical greenness indices. Besides NDVI, which tends to saturate asymptotically under moderate-to-high biomass conditions (Huete et al., 2002; Gitelson et al., 2003), we analyzed SAVI—the index developed to compensate for the reflectance from soil (Huete, 1988), and WDRVI which was developed to increase the linearity with biophysical parameters (Gitelson, 2004). SAVI improved the accuracy of GEPd estimations only for winter rye (NRMSE of SAVI model was smaller by 33% than for NDVI-based model), but did not improve the accuracy of GEPd estimations of neither winter wheat nor spring crops. WDRVI improved estimations of GEPd only for spring barley (NRMSE of WDRVI model was smaller by 16% than for NDVI-based model), and did not affect the GEPd model accuracy of other crops (Table 4). WDRVI explained maximum 62% of GEPd variability for winter rye and only 50% for winter wheat and RMSE in both cases was not very much different from NDVI- or SAVI- based models. In case of spring crops WDRVI performed much better and explained around 85–86% of variability in GEPd for spring barley and potato, respectively, although only for spring barley this index was the best proxy of GEPd (Table 4, Fig. 5). NDVI explained even 91% of GEPd variability of potato being the best predictor of GEPd.

Figure 5 Relationships between the best fitted VIs and GEPd as well as between observed and predicted daily GEP for spring barley (A, B) and potatoes (C, D).

Predicted GEPd was calculated based on VI model resulting in the best goodness of fit (see Table 4).

Inclusion of PAR into VI-based models, although successfully implemented in many studies (e.g., Gitelson et al., 2006; Wu et al., 2009; Rossini et al., 2010), did not improve goodness-of-fit of the linear regressions for any of the crops. Similar results have been reported also by Rossini et al. (2012) and Sakowska et al. (2014) for alpine grassland ecosystems. Sakowska et al. (2014) hypothesized that this might be the result of a different response of plant photosynthesis to direct and diffuse radiation, but this cannot support our results, because all measurements were taken under sunny days. As stated by Rossini et al. (2012), although models including PAR and VI take into account variations related to changing incident radiation, they may not improve estimation of GEP due to higher light use efficiency (LUE) of plants at lower values of incident PAR and lower LUE at higher PAR (due to higher photoinhibition). Rossini et al. (2012) argued that higher photosynthetic efficiency at low PAR may be the result of two processes: (1) more diffuse light is penetrating more deep into the canopy, and (2) less photoinhibition on the top of the canopy which may reduce tendency towards saturation (Chen, Shen & Kato, 2009). It is well known that exposure of photosynthetic machinery of plants to strong light may result in inhibition of the photosystem II (PSII) activity, due to toxic effect of reactive oxygen species (Murata et al., 2007). Although this effect may be overcome by rapid and efficient repair of PSII (Aro, Virgin & Anderson, 1993), environmental factors, such as e.g., heat stress, which often occur during growing season, may inhibit the reparation of PSII and hence reduce efficiency of CO2 uptake (Murata et al., 2007). This is probably why we observed a very weak correlation between GEPd and average daily values of incident PAR (Fig. 6). Although GEPd was increasing with increasing PAR during the vegetative phase of crop development, LUE became stable, while LAI and biomass of plants have been increasing continuously (Figs. 2 and 6). During the reproductive phase, both GEPd and LUE decreased with decreasing values of PARd, but this effect is clearly related to progressive degradation of photosynthetic apparatus towards the senescence. Considering the above, we may hypothesize that the increasing GEPd observed during the vegetative phase of plant development cycle is mostly related to increasing biomass of plants and amount of photosynthetic apparatus rather than to increasing PAR. Another point is that the chamber measurements of CO2 fluxes were taken on sunny days, when PAR was not a limiting factor for GEP. This is probably why we found a week correlation between GEPd and average daily PAR, which can indicate that over such a long seasonal time scales PAR is not the most relevant determinant of GEPd, although in shorter time-scales it may be more important (as indicated also by Sims et al., 2006).

Figure 6 Scatterplots of relationships between average daily PAR (PARd) and GEPd (A) as well as PARd and LUE (B) for vegetative and reproductive phases of plant development.

Data for all the crops are presented on the graphs.

One of the possible explanation, why the simple greenness indices considered in this study were moderately correlated with GEPd of winter crops might be that we considered all the 3-years period dataset together for all the crops, while it is clear that climatological conditions for all these years were extremely different. Especially, 2011- a year with a very dry spring, has disturbed the relationships between GEPd and VIs (mainly for winter crops) and although not shown, the same VIs-based models obtained specifically for 2012 and 2013 resulted in a much higher accuracy. However, our intention was to test the multiyear relationships between VIs and GEPd in order to develop more general models which can be applied to years characterized by different climatological conditions, still maintaining a reliable performance.

Figure 7 Example of seasonal courses of GEPd, NDVI, LAI and NDVI*LAI for winter wheat (A), winter rye (B), spring barley (C) and potato (D) in 2012.

Note: scales are different.

Figure 8 Scatterplots of relationships between NDVI and GEPd as well as between NDVI*LAI and GEPd for the analyzed crops.

(A and B) winter rye; (C and D) winter wheat; (E and F) spring barley; and (G and H) potato. Relationships are determined for vegetative and reproductive phases of plant development cycle.

Another reason for higher uncertainties of GEPd estimations with VIs, observed specifically for winter crops, might be related to seasonal changes in canopy structure and biochemical traits which may modify the spectral response of plants over the growing season at different phenological phases of plant development (Asrar et al., 1984; Asrar, Kanemasu & Yoshida, 1985; Peng et al., 2017). The seasonal courses of GEPd and NDVI are nearly the same and are overlapping for potato (Fig. 7) and that is why the greenness indices (NDVI, SAVI and WDRVI) are good proxies of GEPd for this crop (R2 for GEPd vs. VIs relationships of 0.85, 0.90 and 0.91 for WDRVI, SAVI and NDVI respectively, see Table 4). However, in case of winter crops and spring barley these GEPd and NDVI seasonal courses did not correspond as well as they did for potato. We observed clearly a shift in seasonal courses of NDVI and GEPd for spring barley, winter rye and winter wheat, and peaks of NDVI occurred earlier than peaks of GEPd for all these crops (Fig. 7). It is well known that NDVI saturates at moderate-to-high biomass conditions (Gitelson et al., 2003), but this effect seems to be crop specific and may depend on the structure of the crop canopy. In spring barley NDVI was increasing together with LAI and GEPd until DOY140 (Fig. 7). After this day NDVI saturated and stabilized at around 0.8 until DOY165, although GEPd and LAI were already decreasing. For winter wheat, NDVI also followed LAI development during the vegetative phase (untill DOY140), but until DOY120 the increase rates of NDVI were slower and its value did not exceed 0.4 since the beginning of the growing season, while GEPd increased much faster during this period. Hence, we can hypothesize that GEPd vs. NDVI relationships might be different in the period between DOY80 to DOY120 and between DOY120 to DOY140, although we cannot confirm this due to not sufficient amount of data. After DOY140, NDVI saturated and stabilized again at around 0.8 until DOY165 and just after it begun to decrease slowly and the rate of decreasing was the same as the rate of GEPd decrease. Hence, in the reproductive/senescence phase of wheat development from DOY165 till the harvest, again GEPd vs. NDVI relationships were significantly (p < 0.05) different (Fig. 8). Much more complex analysis are related to winter rye, where NDVI reached maximum values of around 0.9 very soon after the beginning of the analyzed period and was quite stable until DOY140, although LAI and GEPd were continuously increasing until DOY145. When analyzing the winter crop data, specifically winter rye, one should take into account that this crop is sawn in the late September under the climatic conditions of the Central Europe and during warm winters it can continue to grow. In early spring, after the beginning of the growing period, the canopy of the crop may get very dense and green if it is growing under non N-limited conditions. In our case, the winter of 2012 was warmer than in 2011 and 2013, with a very warm December and January 2012 (Fig. 1), hence crops continued to grow during this time. Considering above, this may explain why NDVI saturated already in the late March/beginning of April 2012 at winter rye fields (Fig. 7). In 2011 and 2013 this effect was also observed, but beginning of the period when NDVI started to saturate occurred one month later (in April), due to longer and colder winters. For the first part of the growing season of 2012 for the winter rye, NDVI was not sensitive to changes neither in biomass, nor in GEPd (the same was observed in all the three analyzed years, data shown in (File S7). Between DOY160 and the harvest, changes of NDVI followed changes of GEPd, but again the GEPd vs. NDVI relationships were significantly (p < 0.05) different than those found for vegetative phase (Fig. 8).

Similar kind of hysteresis were found in relationships between GEPd and LAI and GEPd vs. chlorophyll content in maize by Gitelson et al. (2014), as well as between reflectance in red and blue bands and greenness indices (e.g., NDVI) vs. chlorophyll content in maize and soybean by Peng et al. (2017). The relationships between these variables were significantly different for vegetative and reproductive phases of maize and soybean development. In our study we found similar kind of differences between GEPd and NDVI in both analyzed winter crops, but not in spring barley and potato. Each of the crops has different height, LAI, canopy architecture, and different contribution of soil to canopy reflectance. As already indicated e.g., by Gausman et al. (1971) and Gausman, Rodriguez & Richardson (1976), and discussed by Peng et al. (2017), optical properties of leaf and canopies are greatly impacted by leaf structure and canopy architecture and that is why relationships between biophysical properties of crops and VIs are often different at different phenological phases of plant development (Asrar et al., 1984; Asrar, Kanemasu & Yoshida, 1985; Gitelson et al., 2014). In winter crops the canopy is very green, dense and more “closed” at the beginning of the growing season and hence reflectance from soil is minimized, while absorption in the red part of the spectrum is high. That is why NDVI saturates early in the season and it does not change much over the vegetative phase of crop development. Whereas in the reproductive/senescence phases, when the leaf structure and canopy architecture change and chlorophyll content is reduced, more light can penetrate deeper into the canopy and that is why the soil reflectance contribution is also higher. Due to differences in canopy architecture and leaf structure and different patterns of light absorption and reflectance by crops as well as different contribution of soil into overall reflectance of vegetation canopies at different phases of phenological development of plants, the relationships between GEPd and VIs of winter crops are different for vegetative and reproductive phases, as indicated also by (Peng et al., 2017; Gitelson et al., 2014). In potato, canopy is more open since the beginning of the growing season, and soil reflectance contributes significantly to the overall canopy reflectance. With the development of the green biomass, absorption of the red part of the spectrum is increasing, while soil contribution is decreasing. After the peak of biomass, during the reproductive and senescence phases, when chlorophyll pigments are degrading while leaves are folding, more light penetrates deeper to the canopy and again contribution of the soil reflectance to the overall canopy reflectance increases, whereas absorption in the red part of the spectrum decreases. Probably this effect can cause lack of hysteresis in GEPd vs. NDVI relationships for this crop, as the slopes of curves for this relationships in both phases of potato development are the same (Fig. 8).

In winter crops, LAI started to increase since the beginning of the growing season throughout the vegetative phase and GEPd followed changes in the crop biomass, although NDVI was already not sensitive enough to track changes in photosynthesis. The same LAI development was found in spring barley and potato, however NDVI followed changes in LAI and as discussed above, it was sensitive enough to track changes in GEPd of spring crops. After the peak of biomass, LAI of all the crops decreased slightly and stabilized at around 2–3 m2 m−2 until the harvest (note, we are not investigating greenLAI, but total LAI). This led to the conclusion, that LAI can be successfully used to overcome problems with greenness indices, which seem to saturate very early in the growing season in winter crops. By multiplication of NDVI-based VIs by LAI this specific issue of seasonal GEPd vs. VIs relationships can be overcome and model uncertainties can be reduced (see Table 4). The curves presenting the seasonal variations of the NDVI*LAI product are more closely related to seasonal changes in GEPd (Fig. 7) and this effect is not only restricted to winter crops, but can also be observed in case of spring crops. For both winter crops the slopes of the GEPd vs. NDVI relationships are significantly different (p < 0.05) between vegetative and reproductive phases and between the species, which can limit their application for accurate estimation of GEPd of these canopies over the entire season (Fig. 8). However, after multiplying NDVI by LAI slopes for the same relationships are much closer to each other (Fig. 8), and uncertainties of GEPd estimations are lower, as indicated in Table 4. Moreover, this approach does not change significantly slopes of the GEPd vs. NDVI relationships for spring crops (Fig. 8), although uncertainties of GEPd estimations for potato can be higher (Table 4).

However, in the more general, crop-combined dataset representing both winter and spring crops, the multiplication of NDVI and LAI led to an improvement of GEPd estimations (Table 4). NRMSE for the model fed with NDVI*LAI was about 14%, and it explained around 74% of GEPd variability independently from the crop species. The obtained accuracy is in the range which can promote this approach in remote sensing studies in order to overcome the hysteresis of GEPd vs. VIs relationship between vegetative and reproductive phases, which indeed may limit their applicability to predict photosynthesis of different crops over the entire growing season.

The limitation of this approach is that VIs are the remote sensing source of information which can be obtained from space, while space born products of LAI are result of (1) statistical models which quantify the relationships between LAI and canopy reflectance or VIs (Baret & Guyot, 1991; Verrelst et al., 2015; Linker & Gitelson, 2017), or (2) different radiative transfer models (Baret et al., 2007; Richter et al., 2012). These relationships between LAI and VIs are often canopy structure- and land-cover depended and are highly impacted by leaf angle distribution, vegetation clumping, optical properties of leaf and canopies (Goward & Huemmrich, 1992). What is more, different canopies may exhibit large variations in reflectance properties which can result in different values of VIs for similar values of LAI and other biophysical parameters (Pinty, Leprieur & Verstraete, 1993). There are few satellite based products of LAI (Zheng & Moskal, 2009) which are based not only on single vegetation indices such as EVI (Huete et al., 2002) or Reduced Simple Ratio (RSR, Brown et al., 2000), but also on linear or non-liner models including many vegetation indices which are used to estimate and map LAI at the landscape and global levels with Landsat satellites (Cohen, Maierspergerr & Tumer, 2003). Hence, we believe that this kind of data (LAI retrieved from space borne data) can also be used to mitigate the hysteresis effect described above and increase the accuracy of GEPd estimations for croplands. We can even speculate, considering that greenLAI and total LAI analyzed in this study are the same in the vegetative phase of plant development, while greenLAI is decreasing till “zero” in the senescence phase, although the total LAI remains relatively constant during this phase, that the product of NDVI and greenLAI retrieved from satellite data will improve estimations of GEPd of winter crops even better than total LAI used in this study. GreenLAI multiplied by greenness related VIs shall improve model accuracy both in the vegetative and reproductive phases of plant development cycle, due to reasons described above. Hence, we hypothesize that the overall hysteresis effect observed in relationships between greenness related VIs and GEP will be reduced. However, this effect will need to be studied in the future in more detail with the application of both greenLAI estimated at the ground and based on satellites data.

The proposed approach and empirical models developed in this study can be tested in other regions and for other C3 crops in order to verify the validity of our assumptions. Considering that our models were developed based on four different crops and three years characterized by different climatic conditions we believe that application of the proposed approaches and formulas can result in reliable estimation of daily GEP values of crops also for other regions with similar climate and crop management systems.

Conclusions

The analyzed multiyear relationships between GEPd and VIs showed that only in the case of spring crops GEPd can be estimated with a high accuracy and with an error smaller than 12% based on simple greenness indices (NDVI, SAVI, WDRVI). The same kind of analyzes conducted for winter crops are much less accurate, and the error of GEPd estimation is higher than 18%.

The reason for the weaker correlation between daily GEP and VIs of winter crops may be related to hysteresis effect of this relationship found between the vegetative and reproductive phases of plant development cycle. We found that multiplication of greenness indices by LAI (which is much more sensitive to changes in biomass and GEPd of winter crops, specifically during the vegetative phase of their development) can mitigate such effect. The product of multiplication of LAI and VI has in most cases the same seasonality as GEPd and that is why it represents well the seasonal changes of gross CO2 fluxes of croplands.

In order to propose as universal model as possible, we investigated the relationships between VIs and GEPd for the cereals- and crop-combined datasets, where both winter and spring crops, as well as cereals and potato were included. We found that there is no difference in GEPd estimations between these two kind of approaches and this general model based on approach where NDVI is multiplied by LAI, can be successfully applied for both winter and spring crops as well as for cereals and potato, while the error of this estimation is not higher than 14%. However, this approach underestimated GEPd of winter crops and potatoes, and overestimated GEPd of spring barley, but the rate of over-or under-estimations was not higher than 25%.

Supplemental Information

File S1 Crops specific values of GEP, LAI, spectral vegetation indices and their product with PARd and LAI

Click here for additional data file.

File S2 Climate data and crop-specific modeled GEP as well as LAI and vegetation indices

This data are used to prepare Figs. 1 and 2.

Click here for additional data file.

File S3 Raw data used for preparing Fig. 3

Data are provided in the format of STATISTICA, StatSoft

Click here for additional data file.

File S4 Raw data used for preparing Fig. 4

Data are provided in the format of STATISTICA, StatSoft.

Click here for additional data file.

File S5 Raw data used for preparing Fig. 5

Data are provided in the format of STATISTICA, StatSoft.

Click here for additional data file.

File S6 Raw data used for preparing Fig. 6

Data are provided in the format of STATISTICA, StatSoft.

Click here for additional data file.

File S7 Raw data used for preparing Fig. 7

Seasonal courses of GEPd, LAI, NDVI and NDVI*LAI for all crops and years 2011-2013.

Click here for additional data file.

File S8 Raw data used for preparing Fig. 8

Data are provided in the format of STATISTICA, StatSoft.

Click here for additional data file.

The authors sincerely thank Prof. Andrzej Blecharczyk and Dr. Wojciech Waniorek from Agronomy Department (AD) of Poznan University of Life Sciences for their support in implementing our study on fields of the long-term experimental farm of AD in Brody. We would like to thank Jędrzej Nyćkowiak, Klaudia Ziemblińska, Krzysztof Ławiński and Daria Polmańska for their help in CO2 chamber measurements and Anna Binczewska for the help in reflectance and LAI measurements. We would like to thank Dr. Anshu Rastogi for his comments to the manuscript and anonymous reviewers for all valuable suggestions which helped to improve the quality of the revised manuscript.

Additional Information and Declarations

Competing Interests

Author Contributions

Data Availability

The authors declare there are no competing interests.

Radosław Juszczak conceived and designed the experiments, performed the experiments, analyzed the data, contributed reagents/materials/analysis tools, prepared figures and/or tables, authored or reviewed drafts of the paper, approved the final draft.

Bogna Uździcka performed the experiments, analyzed the data, contributed reagents/materials/analysis tools, prepared figures and/or tables, authored or reviewed drafts of the paper.

Marcin Stróżecki performed the experiments, approved the final draft.

Karolina Sakowska performed the experiments, authored or reviewed drafts of the paper, approved the final draft.

The following information was supplied regarding data availability:

The raw data are provided in the Supplemental Files.

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
