# Peer review of "Improving remote estimation of winter crops gross ecosystem production by inclusion of leaf area index in a spectral model"

_PeerJ, doi:10.7717/peerj.5613_

## Round 0.1 · original submission · Major Revisions

Please address the comments from the two reviewers, particularly Reviewer 1.

Reviewer 1 ·

Basic reporting

1.There are most of redundancy contents in discussion section, please reframe the structure and contents of this part (e.g., Line316-345 these contents should appear in the introduction). Meanwhile, authors should discuss the method and analyze specificly why the method you used underestimates GEPd of winter crops and overestimates GEPd of spring crops?

2.L393: using Table4 instead of Table3

3.Figure 2: It is not explicit about the image information. What do triangles, circles and A~E represent, respectively?

4.Figure 6: Why the value of NDVI*LAI for winter rye is higher than GEP at vegetative stage, what about other years? Is it the reason why the coefficient of determination is so low (r2=0.01) in figure 7 (A)?

Experimental design

1. Why EVI is not used and tested but the precursor SAVI is used?

2. Line 203: How did you calculate NDVI? Provide a formula.
Compared with NDVI, EVI is widely used in various models of Gross Ecosystem Production and shows to be sensitive to vegetation growth, why not consider this vegetation indices? If is the method limitation of multispectral radiometers (Line 347-348), why not use remote sensing data?

3. Line207-215: It is not clear how the regression was done? Did you assume that the models output should be equal to the total GEP from measure data and then used this information to calculate variable a and b? If this is the case, how was the data divided between calibration and validation? Maybe you need to add more experimental data.

4. Should the model be calibrated and validated separately by including all crop-combined dataset? There are also problems with the structure of the latter part of article and amount of crop-combined dataset are not enough at daily scale.

Validity of the findings

1. Line153-154 & Line191-192: how did authors acquired the daily data? Please explain more. The data used in all methods are obtained by authors under ideal experiment conditions (Sunny day). Can these methods be applied to in situ measurement data?

2. L373-375: All measurements were taken over the sunny days in this research, so how did you explain the weak correlation between PAR and GEP is caused by long time scales?

3. For all figures, scatter plots data for each crop at daily scale are not enough (less than the amounts of 3 years). Please check your data.

Additional comments

The authors considered the hysteresis of relationship between vegetation index and GEP and propose methods to increase modeling accuracy of daily GEP. The authors intended to propose a general model, but from the results, the general model does not seem working well. I suggest major revisions with specific comments.

Reviewer 2 ·

Basic reporting

Some English editing is required.

Experimental design

The research question is not clearly defined, but only ambiguously implied.

Validity of the findings

No comments. Please see the general comments to the authors.

Additional comments

The study aims to estimate GEP from VI and LAI, which could be retrieved from remote sensing approaches, and suggests that for winter crops, the product of VI and LAI estimates GEP better than VI alone, due to reduced hysteresis of the relationship between GEP and VI. In general, this study provides a feasible approach for routinely estimating GEP of some particular field plots, which could be used for other similar scenarios. So, I recommend this paper be published—when the following concerns and comments are addressed by the authors.

Line 18: the title is kind of too long and can be condensed.
Line 30: “vegetative stage”—this is a little ambiguous.
Line 38: what is “crop-combined dataset?” Please rephrase.
Line 41: it’s counter-intuitive that greenness index saturate in the early growing season.
Line 78: “remote estimate of LAI” – please rephrase.
Line 106: if this model is empirical, can this model be applied to other vegetation types or to same vegetation types in other regions? It would be good if the authors could address this apparent concern in the paper.
Line 147: “flow rate of 0.7 liters” → maybe the unit is not correct?
Line 151: “sunny and cloudless days” → this is repetitive.
Line 239: “dependently on the …” → depending on the…
Line 295: any potential reason for the divergent cases?
Line 373-374: could it be that all the measurements were made on sunny days, so PAR is not a limiting factor for GEP, and hence there is no relationship between PAR and GEP?
Line 384-385: This seems the main purpose of this study. Maybe the authors should state it in the abstract as well as conclusion parts.
Line 467: “what is more” → moreover.
Line 479-480: the statement about satellite remote sensing retrievals of LAI is not accurate.

---

## Round 0.2 · accepted · Accept

Please try to reduce the length of the title to make it more concise.

# Reviewer 1 ·

Basic reporting

Clear and unambiguous

Experimental design

Research question is well defined

Validity of the findings

Data is statistically sound

Additional comments

The author has addressed the questions appropriately. I do not have further comments except that I feel the title is too long. It is best for the author to revise the title to be more concise.

Reviewer 2 ·

Basic reporting

The manuscript has improved significantly after the revision.

Experimental design

The experimental design has improved significantly after the revision.

Validity of the findings

The manuscript has improved significantly after the revision.

Additional comments

The manuscript has improved significantly after the revision, and I recommend this paper be published.